REGISTERED REPORT PROTOCOL

# Epigenetic, psychological, and EEG changes after a 1-week retreat based on mindfulness and compassion for stress reduction in healthy adults: Study protocol of a cross-over randomized controlled trial

**Gustavo G. Diez**[1,2], **Ignacio Martin-Subero**[3], **Rosaria M. Zangri**[4], **Marta Kulis**[3], **Catherine Andreu**[5], **Ivan Blanco**[4], **Pablo Roca**[1,6], **Pablo Cuesta**[7], **Carola García**[1,8], **Jesús Garzón**[1], **Carlos Herradón**[1], **Miguel Riutort**[1], **Shishir Baliyan**[9], **César Venero**[9], **Carmelo Vázquez**[1,4] *

1 Nirakara/Lab, Madrid, Spain, 2 Ph.D. Program in Neuroscience, Department of Anatomy, Histology and Neuroscience, School of Medicine, Universidad Autónoma de Madrid, Madrid, Spain, 3 IDIBAPS, Barcelona, Spain, 4 School of Psychology, Complutense University of Madrid, Madrid, Spain, 5 Polibienestar Institute, University of Valencia, Valencia, Spain, 6 School of Psychology, Villanueva University, Madrid, Spain, 7 School of Medicine, Complutense University of Madrid, Madrid, Spain, 8 Mindfulness Vivendi, Madrid, Spain, 9 School of Psychology, Faculty of Psychology, UNED, Madrid, Spain

* cvazquez@ucm.es

## Abstract

### Introduction

The main objective of the study will be to evaluate the effects of two widely used standardized mindfulness-based programs [Mindfulness-Based Stress Reduction (MBSR) and Compassion Cultivation Training (CCT)], on epigenetic, neurobiological, psychological, and physiological variables.

### Methods

The programs will be offered in an intensive retreat format in a general population sample of healthy volunteer adults. During a 7-day retreat, participants will receive MBSR and CCT in a crossover design where participants complete both programs in random order. After finishing their first 3-day training with one of the two programs, participants will be assigned to the second 3-day training with the second program. The effects of the MBSR and CCT programs, and their combination, will be measured by epigenetic changes (i.e., DNA methylation biomarkers), neurobiological and psychophysiological measures (i.e., EEG resting state, EKG, respiration patterns, and diurnal cortisol slopes), self-report questionnaires belonging to different psychological domains (i.e., mindfulness, compassion, well-being, distress, and general functioning), and stress tasks (i.e., an Arithmetic Stress Test and the retrieval of negative autobiographical memories). These measures will be collected from both groups on the mornings of day 1 (pre-program), day 4 (after finishing the first program

OPEN ACCESS

**Data Availability Statement:** All relevant data from this study will be made available upon study completion. This paper only reports a research protocol but we include now an OSF link with all the the questionnaires of the protocol and the Informemd consent.

**Funding:** CV: Ministry of Science PID2019-108711GB-I00 RMZ:Spanish Ministry of Science FPI predoctoral fellowship (PRE2020-092011) GD: Mindfulness and Cognitive Science Chair of Complutense University of Madrid. The funders did not and will not have a role in study design, data collection and analysis, decision to publish, or preparation of the manuscript.

**Competing interests:** The authors have declared that no competing interests exist.

and before beginning the second program), and day 7 (post-second program). We will conduct a 3-month and a 12-month follow-up using only the set of self-report measures.

## Discussion

This study aims to shed light on the neurobiological and psychological mechanisms linked to meditation and compassion in the general population. The protocol was registered at clinicaltrials.gov (Identifier: NCT05516355; August 23, 2022).

## 1. Introduction

In recent years there has been an outburst of scientific studies on the positive effects of mindfulness-based interventions (Mindfulness-Based Interventions, MBIs) on physical and mental health [1]. Two of the most researched programs related to the development of mindfulness and compassion are, respectively, Mindfulness-Based Stress Reduction (MBSR) [2], focused on attentional training and a non-judgmental attitude, and Compassion Cultivation Training (CTT) [3, 4], more focused on the structured training of a compassionate and self-compassionate attitude. These are highly structured programs, of short duration (i.e., typically delivered in 8 weeks/16 hours format), that have shown positive effects, both for general and clinical populations, as reflected in many meta-analyses and systematic reviews [5–9], including programs using briefer formats (e.g., single-session to 2-week multi-session formats) [10].

The fundamental objective of this study is to explore the potential changes in epigenetic variables, psychological distress, well-being, brain activity, cortisol levels, and reactivity to stress, that can be observed following MBIs. The way to address this question is through an intensive intervention in a retreat format, a procedure that allows control of important variables (e.g., food, environmental conditions, etc.), and administering all the components of the standard MBSR and CCT programs in their entirety. There is already meta-analytic evidence on the positive effects of these intensive multi-day programs in healthy individuals [11]. Although there is some evidence that attentional changes might be involved in the psychological changes observed after a meditation retreat [12], there is a need to examine the mechanisms of action through which meditation practice produces its effects [13]. Most theoretical models published to date emphasize the central role of attention regulation, which is thought to underpin emotional and cognitive flexibility [14], which in turn enhances emotion regulation processes [15], and the ability to maintain non-judgmental awareness of thoughts, feelings, and experiences. Theoretical models also emphasize the role of compassion and self-compassion [16] in promoting change due to MBI practices [17–21]. However, the extant evidence comparing the self-reported psychological effects of MBSR and CCT programs indicates that both programs have similar effects [22, 23], although reached through different mechanisms [24].

Regarding the biological effects of MBIs, there is consistent meta-analytic evidence showing that they produce significant physiological changes in parameters like a reduction in oxygen consumption, blood pressure, and heart rate [25], which are contrary to those that occur during the stress response [26]. These changes have also been observed in three-day intensive meditation retreats [27]. Similarly, although the evidence for brain volumetric change is uncertain [28], there is data on changes in frontoparietal and default network connectivity during resting state tests [29, 30]. There is also promising evidence that intensive intervention practices, in a retreat format, are associated with significant changes in gene expression that might be potentially regulated by epigenetic mechanisms. For example, an 8-day silent guided

meditation retreat can modify gene expression, in particular genes related to the immune response [31]. However, the brain mechanisms and molecular activity are not yet well understood. Along the lines of some previous research on detectable epigenetic changes in brief intensive meditation interventions aimed at reducing stress [32–36], we intend to test whether epigenetic changes are one of the mediating mechanisms in the observable psychological changes following an intervention. We will also analyze whether changes in brain connectivity (measured with EEG in resting state) and in the diurnal cortisol rhythms occur throughout the programs.

Although research on the comparative effects of MBSR and CCT has shown the existence of some particular mediators [24], the effects of a sequential intervention of both types of training on the same individuals are not known. This sequential approach is important to evaluate whether there are additive effects of the programs and whether the order of the intervention (one more focused on attentional aspects and the other more on socio-emotional factors) can facilitate psychological and neurobiological changes. Thus, we aim to analyze the existence of differential psychobiological changes in two types of intensive psychological programs (MBSR and CCT), and whether the sequence of the programs (using a cross-over design), does significantly affect the results.

As, to the best of our knowledge, this is the first study using a sequential design with two intensive meditation modalities, it is difficult to formulate specific hypotheses regarding the separate and combined effects of the MBSR and CCT programs on the selected outcomes. Thus, considering the two MBIs jointly, we expect that both programs will significantly: a) modify the DNA methylation and expression levels of specific genes and a reduction of the DNA methylation-based biological clock; b) reduce participants' levels of psychological distress while increasing their levels of psychological functioning (i.e., mindfulness, compassion, emotion regulation, and well-being); c) change the EEG spectral profile of the resting state brain activity (i.e., changes in the balance between Alpha (8–14 Hz) and Theta (4–8 Hz) frequency bands and, regarding the organization of the brain functional networks, a modulation of the difference between the activation of the default mode network and the one corresponding to the salience network); d) alter awakening and bedtime cortisol levels leading to a steeper Diurnal Cortisol Slope (DCS); e) improve the efficacy of mood regulation following a negative mood induction (using an autobiographical memory task); f) reduce participants' reactivity in an objective experimental stress test (i.e., Arithmetic Stress Test). In addition, given the multidisciplinary nature of the present project, we intend to provide a comprehensive view of potential changes in various areas of biological and psychological functioning and their interrelationships.

In general, we expect an overall pattern of results in which changes will be significantly higher for the combined effects of both MBIs than for each program separately. Moreover, based on previous literature comparing standard MBSR and CCT programs [24], we expect no major differences in primary outcomes between both programs, although the mechanisms of change may be different; in general, changes in MBSR will be mediated by cognitive variables (e.g., mindfulness) whereas changes in CCT will be mediated by socio-emotional variables (e.g., compassion). We also expect that the self-reported psychological changes associated with the MBIs will remain 3 months after the programs.

## 2. Methods

### 2.1 Study design overview

We used the SPIRIT reporting guidelines. Participants will be randomly assigned to two groups starting with MBSR or CCT. A stratified randomization procedure, using the Research

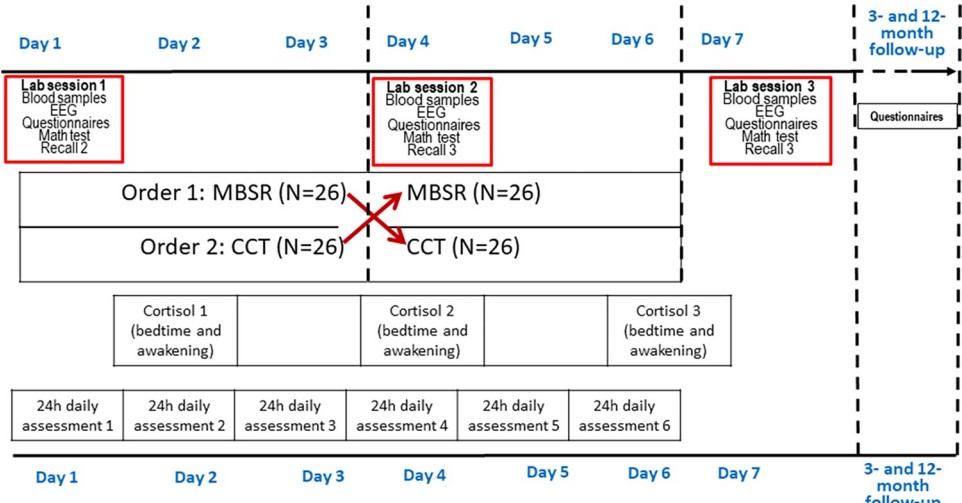

**Fig 1. Cross-over design.**

Randomizer Program (https://www.randomizer.org), performed by the principal investigator (CV), will be used to maintain the same percentage of male and female participants in each group. Participants from both groups will take part in a 7-day retreat, including training in both meditation-based practices. Participants will be blinded to group allocation starting with MBSR or CCT and will not receive any information of any kind about the study hypotheses. (IB and RZ will be in charge of participant recruitment and perform the randomization of each study group). Using a cross-over design, the participants will undergo their second training during the last three days of the retreat. Biological and psychological measures will be collected from both groups in the mornings of day 1 (before beginning the programs), day 4 (after the first program and before the beginning of the second program), and day 7 (at the end of the program). All psychological questionnaires will be administered online. Also, before going to bed, participants will be given a short questionnaire with eight questions about their general functioning during the day and their satisfaction with the progress of the program. Furthermore, a 3-month and a 12-month follow-up assessment will be conducted in both groups only for psychological questionnaires. (See Fig 1).

## 2.2 Participants: Recruitment and selection

Individuals from the general population will be recruited through the Nirakara-Lab website, a university-associated research center specializing in Mindfulness and Compassion-Based programs for the general population. Participants will be invited to join the study on the university's official website offering the MBSR and CCT programs (www.nirakara.com). This is a widely known national website for the general public interested in research and training in meditation. An e-mail will be sent to the thousands of individuals registered in the website newsletter. In any case, participation in the study will be open to any interested individual who meets the inclusion and exclusion conditions. To control for potential self-selection biases, which is a common shortcoming of MBIs research [37–39], there will be statistical controls on key issues such as years and type of previous meditation practice, if any, sociodemographic information, and mental health status. Individuals interested in participating in the retreat will be interviewed by members of the research team (RZ and CA) to screen their suitability for the study. Inclusion criteria will be being a healthy adult between 25 and 65 years old who is

interested in meditation and compassion techniques to regulate their mood and stress. Exclusion criteria include having a current or past self-reported diagnosable serious or disabling mental disorder (in particular, PTSD, major depression, psychotic disorders, and/ or use of alcohol or drugs disorders) or having a current or past (i.e., less than 5 years ago) medical or physical condition that may affect the immune system (i.e., autoimmune disease, chronic severe infections, HIV, cancer), pregnancy, serious chronic illnesses, as well as the consumption of psychotropic drugs. Individuals with current habits (i.e., smoking, alcoholism, substance abuse) or taking specific medications (e.g., corticoids) that might affect the immune system are also excluded. Also, individuals planning to travel from a different time zone or long-travel times will be excluded as that might affect the immune system.

The screening procedure will be conducted as follows: 1) First contact will be made through an online form expressing interest and commitment to participating in the study, where participants will be provided with some information on the retreat (date, location, and program characteristics) and will complete information on demographics, physical and psychological health history, and meditation experience. A waiting list will be formed for participants who did not sign up in time and wished to be considered in case a place became available. An expected final sample of 52 people will take part in the retreat being randomised to the MBSR training or CCT training for the first part of the retreat. All participants will complete the first arm of the study and then, after a half-day break, will be allocated to the opposite condition for the second study arm (following a cross-over design). There will not be a longer wash-out period due to the difficulties of extending the participants' stay in the retreat for more days. However, in psychological studies with a cross-over design (e.g., [40]) there are usually no long periods between treatments. Moreover, introducing a wash-out period (days, weeks, or months) would introduce difficult-to-control confounders (e.g., different life events experienced by individuals) and would make comparability of treatments more difficult.

Participants will pay the cost of the retreat (i.e., meals and accommodation only) and will not receive any economic compensation for their contribution to the study. Participation requires informed consent, provided by the principal investigator (PI) or authorized full members of the research team, in which all the conditions of the study are reported. In case of any adverse effect, which is very unlikely in this type of intervention, participants may leave the study at any time without having to offer any explanation and without any kind of penalty. Furthermore, the team will facilitate any kind of help requested by participants during the retreat. The collection of biological information is innocuous, and blood and saliva samples will be collected by nursing professionals. Participants will be allowed to drop out at any point during the study as stated in the informed consent.

The study has been approved by the University Ethics Committee (Ref. 22/449-E, July 2022) and has been pre-registered (clinicaltrials.gov, Identifier: #NCT (NCT05516355). Any modification to the protocol will be reported both to the Ethics Committee and the trial registration site. Participants will be informed in advance of the results of the study to be published in scientific journals.

## 2.3 Sample size

According to G*Power software, we need 21 participants per group to obtain a power of .95 to detect a mean effect size of .37 (based on well-being outcomes in mindfulness retreats, [11]), using order of intervention as a between-subject factor (MBSR first vs CCT first) and time as a within-subjects factor (5 times of assessment), with an alpha of .05, a power 95%, and a 2-tailed test. In anticipation that there may be last-minute dropouts in pre-registrants or incomplete data in the sample collection or the analysis of the planned psychological or biological data, we

have increased the sample by an additional 20% to 26 participants for each condition (52 in total). This sample size further corresponds to the average size of standard meditation retreats [11, 41] and in studies of brain activity in meditators [29, 30].

## 2.4 Components and structure of the programs

Each program will comprise standard MBSR and CCT protocols in an intensive format to fit a retreat mode. Throughout the retreat, participants will be in silence, not talking to each other or using electronic devices. Each program will last 3 days in total, and all participants will go through both programs. On each retreat day, participants will spend a total of 4.25 hours of sitting meditation, 2 hours of walking meditation, 2 hours of explanations and discussing with the instructors, doubts or problems related to the practices, and 1 hour of mindful body movements (yoga/qigong). In total, participants will have 9 hours of practice each day. (A description of the contents of the programs is shown in Table 1).

**2.4.1 Mindfulness-Based Stress Reduction (MBSR).** The MBSR is an 8-week standardized program [42] that aims to cultivate awareness by paying attention to the present moment with acceptance and a non-judgmental disposition. Training is delivered by two certified instructors at the University of Massachusetts Centre for Mindfulness (**Error! Hyperlink reference not valid.** www.umassmed.edu/cfm/) and Mindfulness Center at Brown University (https://www.brown.edu/public-health/mindfulness/home).

**Table 1. Description of the main practices of the retreat.**

| Practice | Description | Protocol | Day |
|---|---|---|---|
| **Basics of Mindfulness** | Practices and insights into mindfulness and acceptance. In this part, explanations are given on how to approach practices during the retreat, and how to deal with distractions, unpleasant bodily experiences, or difficult emotional states. | MBSR / CCT | 1 |
| **Yoga-Qigong** | Simple body exercises to relax tensions, unblock joint patterns, and develop mindfulness in movement. In the case of CTT, references are also made to self-care and self-compassion in movement. | MBSR / CCT | 1 |
| **Walking meditation** | The participant is trained in the ability to be aware while walking. The movement is slower than usual. Breathing and walking movements are integrated. In the MBSR protocol, awareness of the breath and the body is encouraged. In CCT, loving-kindness or compassion is developed while walking. | MBSR / CCT | 2 |
| **Body Scan** | Attention is brought to the various parts of the body. From the soles of the feet to the top of the head. The aim is to develop mindfulness. Every time there is a distraction, the attention is gently taken up again on the sensations of the body. | MBSR | 1 |
| **Awareness of Breathing** | Attention is focused on breathing. Two options are given, depending on the preference of the participants (i.e., focusing attention on the tactile sensations of air in the nostrils or on the sensation of the movement of the abdomen). | MBSR | 2 |
| **Awareness of Thoughts** | It aims to focus attention on the activity of the mind (i.e., on the spontaneous generation of thoughts and emotions). | MBSR | 3 |
| **Open Monitoring** | It is about accounting for the constant flow of subjective phenomena: cognition, perception, and emotion. Rather than choosing an object to focus attention, participants are encouraged to engage in a choiceless awareness in which subjective experience is continuously monitored. | MBSR | 3 |
| **Loving-kindness and compassion for a loved one** | Through visualization exercises of loved ones, practitioners are helped to recognize feelings of love whenever they arise. | CCT | 1 |
| **Self-Compassion** | Practising through assertions generates a feeling of self-acceptance and a desire to overcome suffering. | CCT | 2 |
| **Loving-kindness for one-self** | Meditation practices are aimed at teaching appreciation and gratitude for oneself through visualization and verbalization. | CCT | 2 |
| **Common humanity** | Through visualization practices, a sense of interconnectedness is developed (i.e., the knowledge that experiences of suffering are inherent to all human beings, as well as the pursuit of happiness). | CCT | 2 |
| **Compassion for others** | Through visualizations and verbalizations, it aims to develop compassion for others. Starting with a loved one, continuing with neutral people, and culminating with people with whom one has difficult relationships. | CCT | 3 |
| **Active compassion practice** | A practice known as *tonglen* in Tibetan Buddhism, it aims to develop a sense of compassion towards others. The participant visualizes how he or she can reduce suffering in other people by doing something beneficial for them. | CCT | 3 |

**2.4.2 Compassion Cultivation Training (CCT).** The CCT is an 8-week standardized program [3, 43] aimed at cultivating compassion and empathy toward oneself and others, consisting of daily formal and informal practices. Training is delivered by two certified instructors at the Compassion Institute and Nirakara-Lab.

## 2.5 Study outcomes

**2.5.1 Epigenetic changes.** Samples of peripheral blood will be obtained and processed in situ, during the retreat. Participants' blood samples will be collected in two standard tubes (10ml each) that contain EDTA as an anticoagulant. Blood samples will be processed immediately using the standardized Ficoll gradient centrifugation protocol to isolate mononuclear cells. These cells are cryopreserved directly (with 10% DMSO culture medium) and stored appropriately (in a -80 C freezer for short periods and liquid nitrogen for longer storage). Nucleic acids will be extracted in the laboratory of the Biomedical Epigenetics group, IDIBAPS (Barcelona), using DNA and RNA extraction kits. Once the nucleic acids are obtained, they will be stored in suitable freezers (-20 or -80˚C) until their use for molecular profiling techniques, such as DNA methylation arrays (Infinium Methylation EPIC Bead Chip from Illumina, herein called "EPIC methylation arrays"), and gene expression by massive RNA sequencing (RNA-seq). (See Table 2).

**2.5.2 Psychological measures.** *a) Self-report measures*. Most of these measures were developed in studies with the general population and have been also used in previous MBI studies in healthy individuals [22]. They have shown good or excellent psychometric properties and cover selected psychological distress and functioning constructs (i.e., stress, anxiety, depression, mindfulness, compassion, well-being, and emotion regulation) and all will be administered online via Qualtrics with individualized links (Table 2):

1. *Psychological distress*. Depression Anxiety Stress Scales (DASS-21) [44, 45].

2. *Psychological well-being*. Satisfaction with Life Scale (SWLS) [46, 47].

3. *Mindfulness*. State Mindfulness Scale (SMS) [48].

4. *Anxiety*. State-Trait Anxiety Inventory (STAI-20) [49, 50].

5. *Somatic Symptoms*. Patient Health Questionnaire (PHQ-15) [51, 52].

6. *Sleep difficulties*. DSM-5 sleep difficulties [53].

7. *Compassion*. State self-compassion State Self-compassion Scale (SSCS) [54, 55].

8. *Fear of Compassion*. The Yourself subscale from the Fear of Compassion Scale [56].

9. *Emotion Regulation*. State Difficulties in Emotion Regulation (S-DERS-21) [57, 58].

10. *Current mood*. Positive and Negative Affect Schedules (PANAS) [59, 60].

11. *Daily assessments of psychological functioning*. A 7-item scale to assess participants' average state of mindfulness, mind wandering, compassion to others, self-compassion, well-being, the richness of life, and utility of the contents of the program every day.

12. *Affect in the last 3 days*. Hedonic and Arousal Affect Scale (HAAS) [61].

13. *Meditation Adverse Effects*. Meditation-Related Adverse Effects Scale–Mindfulness-Based Program (MRAESMBP) [62].

14. *Program satisfaction*. Client Satisfaction Questionnaire (CSQ-8) [63, 64].

**Table 2. Study procedures at specific time points.**

| | Home baseline screening | Day 1st (BL) | Day 4th | Day 7th | 3- and 12-month follow-up |
|---|---|---|---|---|---|
| | | **7-day retreat** | | | |
| *Epigenetic measures* | | | | | |
| • DNA methylation levels | | x | x | x | |
| • Proportion of main blood cell types | | x | x | x | |
| • Epigenetic clocks | | x | x | x | |
| *Psychological measures (outcomes of programs)* | | | | | |
| • Perceived Stress Scale | x | | | | |
| • Depression Anxiety Stress Scales (DASS-21) | | x | x | x | x |
| • Patient Health Questionnaire (PHQ-15) | x | | | x | x |
| • Patient Health Questionnaire (PHQ-9) | x | | | x | x |
| • Satisfaction with Life Scale (SWLS) | x | | | x | x |
| • State Difficulties in Emotion Regulation (S-DERS-21) | | x | x | x | x |
| • State Mindfulness Scale (SMS) | | x | x | x | x |
| • State Self-Compassion Scale (SSCS) | | x | x | x | x |
| • Fear of Compassion (FOC) | | x | x | x | x |
| • Hedonic and Arousal Affect Scale (HAAS) | | x | x | x | x |
| • Meditation-Related Adverse Effects Scale | | | x | x | x |
| *Psychological tasks* | | | | | |
| • Arithmetic Stress Test | | x | x | x | |
| • Autobiographical memory retrieval | | x | x | x | |
| • Positive and Negative Affect Schedules (PANAS—state) | | x | x | x | |
| • State-Trait Anxiety Inventory (STAI-20) | | x | x | x | |
| *Psychophysiological laboratory* | | | | | |
| • EEG recording (open- and closed-eyes) | | x | x | x | |
| • EKG recording | | x | x | x | |
| • Respiratory rate | | x | x | x | |
| *Cortisol activity* | | | | | |
| • Saliva samples (bedtime and awakening) | | x | x | x | |

Note: BL = Baseline

*b) Psychological stress tasks.* **b.1) Stress Reactivity.** Arithmetic Stress Test: a standardized laboratory stress induction procedure, that is part of the Trier Social Stress Test [65, 66], will be used to assess participants' stress reactivity. Individuals are asked to repeatedly subtract a given 2-digit number (i.e., 13) from a 4-digit number (e.g., 1022). These numbers were changed for each of the three assessment days. Participants are asked to write their responses on a personal computer as quickly as possible. Time is limited to 3 minutes to increase the stressful nature of the task. Pre-post stress level changes in mood will be assessed with the PANAS and STAI.

**b.2) Emotion regulation.** A 10-min retrieval of autobiographical negative memories. Participants will receive a link through Qualtrics for a baseline assessment (three days before the retreat) where they are asked to think about three negative autobiographical memories of events for which they felt responsible and made them feel negative emotions such as shame, guilt, and sadness. Participants will be asked to assign personal keywords (e.g., 'traffic accident'), for each of the three memories, synthesizing the event. Each of the three negative events will be randomly assigned to the three respective assessment days so that participants were given the corresponding "keywords" when they begin the retrieval task. Participants are given

10 minutes to write down the memory as specifically as possible. Pre-post changes in their sadness, happiness, anxiety, serenity, shame, and guilt affect states will be assessed using a 0 (nothing) to 100 (very much) visual analog scale. These two laboratory psychological tasks will be completed by each individual while alone in a quiet room, using a portable computer.

**2.5.3 Neurobiological laboratory measures.** *a) Brain electrical activity*. Recording of EEG activity in resting state (alternating open and closed eyes, 6 min each condition) with four EE-225 64chEEG+24chBIP 16kHz devices with a laptop and eego EEG recording software. The sampling rate will be set at 1000 Hz and the reference is 7Z of the equidistant layout.

*b) Respiratory activity*. Respiration rate, which is related to attention training and psychological well-being [67], will be monitored by a respiration chest-band, during the EEG assessment session and will be registered using a piezoelectric sensor.

*c) Cardiac activity*. EKG activity will be monitored with dedicated electrodes during the EEG assessment session. Based on these recordings, we will analyze changes in EKG, heart rate, and heart rate variability (HRV) patterns

These recordings will be completed by everyone in laboratory spaces dedicated to the EEG assessments.

**2.5.4 Salivary sampling and cortisol measurement.** Participants will be given Salivette® cortisol saliva sample collection tubes (Sarstedt, Germany) that contain a sterile cotton swab for sample absorption along with detailed verbal and written instructions concerning sample collection. Diurnal Cortisol Slope and cortisol levels at bedtime and awakening will be measured at three different time points of the retreat (see Fig 1). Then, the saliva samples will be stored in the refrigerator until they are delivered to the laboratory after a few hours. Once in the laboratory, the samples will be centrifuged at 3000 rpm for 5 min, resulting in around 1.5–2.0 ml clear supernatant of low viscosity that will be stored at −80˚C until the analyses of the salivary cortisol levels. Cortisol, the principal stress glucocorticoid produced by the hypothalamic-pituitary-adrenal axis (HPA) [68], will be assessed using a commercially available enzyme-linked immunosorbent assay (Salimetrics®) having an approximate sensitivity of <0.007 μg/dL.

Table 2 shows a description of all measures and time points.

## 2.6 Procedure

In arriving at the retreat centre, everyone will be informed about the retreatment rules (e.g., accommodation, silence rules, use of electronic devices, etc.) as well as his/her specific time schedules for each of the laboratory sessions. All laboratory sessions will be done on the morning of each assessment day. We will set up different laboratories for the tasks (one for blood extractions, one with nine computers to complete the self-report questionnaires, four EEG separate labs with portable equipment, and another four separate labs for the psychological tasks with one computer per room). The schedule of the assessments will be kept the same for each participant on the three assessment days. The order of the tasks will be the same for all participants (i.e., blood extraction, self-report questionnaires, EEG recording, autobiographical memory task, and arithmetic stress task). The expected total duration of each assessment session is approximately 90 minutes.

## 2.7 Statistical analysis plan

Independent t-tests for continuous data and chi-square test ($\chi$2) for categorical data will be conducted to confirm that there were no demographic or clinical differences between groups at baseline. Repeated-measures ANOVAs will be conducted to analyse changes associated with the programs. For the autobiographical memory task and all the biological measures (i.e.,

genetic measures, EEG, EKG, respiratory rate, and cortisol in saliva), a 2 (Order of program) x 3 (Time) repeated measures ANOVA will be conducted. For the self-report psychological measures, a 2 (Order of program) x 4 (Time) ANOVA will be conducted. We will use age, sex, and previous meditation experience as covariates in the main analyses. Assumptions of normality, homogeneity of variances, sphericity, and homogeneity of covariances will be checked. Further analyses will include post-hoc, p-corrected comparisons (e.g., Bonferroni). All analyses will be performed with SPSS 28.0 (IBM Corp., 2021) or R Studio statistical software (R Core Team, 2021). Two-tailed tests will be used to determine significance at the 5% level.

The EPIC methylation arrays´ raw data will be normalised and filtered by a robust bioinformatics pipeline procedure [69]. We will then perform an unsupervised analysis of the data via hierarchical clustering, i.e., PCA and t-SNE (by R Studio). In parallel, taking advantage of the known DNA methylation signatures related to distinct cell types or molecular processes, we will: 1) characterize the proportion of different immune cells, and 2) determine the epigenetic age with Horvath's model or other epigenetic clocks to capture the influence of lifestyle on the DNA methylome [70, 71]. This may reveal a link between the MBI programs and the immune response, inflammation, and cells' physiological condition. Finally, we will focus on supervised analyses to identify differential DNA methylation biomarkers in the two groups (MBRS first *vs* CCT first) and time points: day 1(baseline), day 4 (program switch), and day 7 (post retreat).

The EEG signals will be assessed both at the sensor level and the source's space. Source space reconstruction would be performed using the eLoretta procedure over the MNI template. The analysis will be conducted two-fold: 1) the power spectral profiles of each condition (eyes closed and eyes open) will be computed using the fast Fourier transform algorithm; 2) the functional networks will be calculated using a phase synchronization algorithm (i.e., phase locking value). The functional networks will be created using standard regions of interest defined over the AAL atlas.

Diurnal Cortisol Slope (DCS) will be calculated individually for each participant as a function of their respective bedtime and awakening times. DCS will be calculated as awakening cortisol subtracted from bedtime cortisol, divided by each subject's total waking hours.

Intention-to-treat (ITT) analyses will be conducted with all participants, regardless of session or outcome measure completion. The pattern and percentage of missing data will be explored before analysing the data. It should be noted that the chances of missing data will be minimized because the questionnaire administration format via Qualtrics does not allow progress to be made until all items in each questionnaire are completed. Binary logistic regression will be used to evaluate the assumption that data is Missing at Random (MAR). ITT mixed models (restricted maximum likelihood (REML) estimation) will be used to account for missing data MAR. If missing were not random (MNAR), we would conduct Multiple Imputation and Pattern Mixture Models incorporating the predictor responsible for the missingness into the model. In any case, sensitivity analysis will be performed, comparing the main outcomes between completers and imputed values. To safeguard the confidentiality of participants' data, all samples will be anonymized with alpha-numeric codes. These codes will not carry any personal information. Personal data linked to the alpha-numeric code will not be included in any files used for data analyses and will remain separate from the main database. Data displayed in public repositories (e.g., GitHub or datadryad.org), linked to the resulting publications, will always be anonymized. Data supporting each published study will be included within the article and/or supporting materials. To improve blinding integrity [72], personnel who will analyse the data collected from the study are not aware of the treatment applied to any given group. The authors will be fully responsible for handling and analyzing data independently from any sponsor. All the questionnaires included in the protocol as well as a copy of the informed consent, are available at https://osf.io/gczas/?view_only=7864451a24cf4a82824f6f298b1bbf0f.

## 3. Discussion

With this study, we aim to shed light on two main aspects. Firstly, we will analyse the precise interaction between psychological and psychobiological changes associated with MBSR and CCT programs. Secondly, following the rationale clinical trials that have compared the effect of the order of administration of two different modalities of interventions [40, 73], we will conduct a cross-over design to explore whether it is important the order of intervention of mindfulness and compassion practices.

This is an ambitious study that attempts to analyze the effects of two meditation programs (MBSR and CCT) by integrating multidisciplinary perspectives. The study will offer a unique insight into the effects on epigenetic variables, using the most advanced analysis methods in the field [69]. Also, adding to current knowledge in the field [74], we aim to correlate epigenetic changes with changes in electrical brain activity and psychological variables, comparing the results before and after the interventions. Thus, we believe that this study offers a great possibility of data integration at various levels that is not common in this type of research. Furthermore, the study has been carefully designed to try to minimize potential biases that are common in meditation research [38, 75]. In this regard, the study employs methods of randomization, selection of well-validated measures, and blinding of statistical analyses.

However, we recognize that the protocol has some limitations. Participants in mindfulness studies are typically self-selected, which is a difficult problem to control in research on meditation procedures. Although the cross-over design will allow us to answer important research questions (e.g., the existence of additive effects of the two modalities), it is not certain that the results can be easily transferable to other settings. Also, the retreats include compact and intensive training that is offered in a short period of time, lacking homework exercises. However, although more time may be required to produce lasting psychobiological changes, previous studies support the idea that brief retreat interventions can detect change [35, 76]. Even in an 8-hour intensive mindfulness-based retreat, significant transcriptional [36] as well as DNA methylation changes were observed in blood samples, and affected in genes related to important cellular pathways (such as fatty acid metabolism, DNA repair, and chromatin remodeling, among others) [35]. Furthermore, differentially methylated CpGs were enriched in binding sites of particular transcription factors, involved in immune response and inflammation [35]. These findings suggest that the changes observed are not random, and seem to be induced by the meditation practice, even after only one day of intervention.

Although we do not know a priori whether we will be able to detect highly relevant epigenetic changes in the blood of the participants of our retreat, there is sufficient published evidence to expect some degree of epigenetic modulation after short periods of practice, supporting thus our experimental design. The same can be said about expected findings in self-report psychological measures. Although the retreat is to last one week, which is the most common duration for meditation retreats [41], it includes the standard contents of the MBSR and CCT programs. Thus, it seems plausible to find the expected significant effects associated with these intensive interventions in that relatively short period. Finally, the influence of the exposure to other contingencies associated with the retreat (e.g., silence, rest days, etc.) may contribute to the changes that may be found [77–80]. We also recognize the limitation that the follow-up is restricted to psychological measures only. Apart from the enormous economic cost of repeating the epigenetic measures, which we cannot afford with our limited funds, it would be impossible to achieve as the participants live in different parts of the country and were recruited only for a one-week retreat.

We hypothesize that even in these very short interventions, we will be able to detect overall changes in meditation-related variables and well-being on both self-report and biological

measures. Incidentally, by examining both psychological and psychobiological outcomes, our study aims to contribute a more holistic understanding of the transformative effects of mindfulness and compassion practices.

To summarize, this is an innovative study aimed at testing the effect of two standard meditation interventions (MBSR and CCT), administered in an intensive retreat format. There is an increasing need for interventions to improve the well-being of the population in a variety of formats. The administration of meditation programs in a relatively short retreat may be an important contribution to meditation studies in the general healthy population. In addition, the cross-over design of the study will provide insight into whether the order of the interventions (MBSR first or CCT first) may play a role in the delivery of these interventions. The design may also help to answer some basic questions in meditation research regarding the ingredients and sequence of the components of these interventions. Thus, our study could pave the way for more nuanced research into meditation-based interventions, especially in terms of optimizing intervention sequences for maximal benefit.

Given the increasing popularity of mindfulness and compassion practices and research in the general population, it is crucial to understand their effects on individuals who do not necessarily have clinical problems but seek personal growth, stress reduction, or well-being enhancement [5, 81–83]. We hope that the rigorous approach of this study ensures the integrity of our findings, free from potential confounders related to health conditions. By understanding the effects of two consecutive interventions on a healthy cohort, we might offer evidence-based guidance on how these practices can serve as preventative measures for the broader community, potentially enhancing its well-being, and perhaps reducing the onset of mental health conditions, We also hope that, despite the potential limitations acknowledged in our design, the results generated by this ambitious study may shed light on the neurobiological and psychological mechanisms linked to meditation and compassion in the general population and may contribute to improving the scientific basis of these practices.

## Supporting information

**S1 File.**
(DOCX)

## Acknowledgments

We thank Nirakara Lab's staff, Ignacio Jiménez and Claudia Caldera for their help throughout the project and all the personnel who will help us to administer the tasks and tests in our labs (Leyre Castillejo, Brenda Nadia Chino, Claudia Cogollos, Alberto del Cerro, Elena Diaz, Lucía Hernández, Marina Izquierdo, Ana Obeso, Ana Mar Pacheco, Ester Santamaria, Lucia Torres), as well as Joaquin Martinez, Head of the Hematology Department of the Hospital Universitario 12 de Octubre (Madrid, Spain) for letting us use his laboratory space, and his technicians Alicia Giménez, Laura Carneros Blanco and Lucía Sastre for their agreement to process biological samples. The support of Miguel Angel Casermeiro (Fundación General UCM) was also very valuable. Special thanks to Yago Piedra, director of the Real Colegio Maria Cristina (El Escorial, Madrid), and all his staff, for their support in preparing the retreat. We also thank Frank Zanow (ANT Neuro, Berlin, Germany) who kindly offered his help to provide technical support for the EEG recordings.

## Author Contributions

**Conceptualization:** Gustavo G. Diez, Ignacio Martin-Subero, Rosaria M. Zangri, Marta Kulis, Catherine Andreu, Ivan Blanco, Shishir Baliyan, César Venero, Carmelo Vázquez.

**Investigation:** Rosaria M. Zangri, Pablo Roca, Pablo Cuesta, Carola García, Jesús Garzón, Carlos Herradón, Miguel Riutort, Shishir Baliyan.

**Methodology:** Marta Kulis, Pablo Roca.

**Resources:** Ivan Blanco.

**Supervision:** Carmelo Vázquez.

**Writing – original draft:** Gustavo G. Diez, Rosaria M. Zangri, Carmelo Vázquez.

**Writing – review & editing:** Gustavo G. Diez, Ignacio Martin-Subero, Rosaria M. Zangri, Marta Kulis, Catherine Andreu, Ivan Blanco, Pablo Roca, Pablo Cuesta, Shishir Baliyan, César Venero, Carmelo Vázquez.

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
