## [Decision Letter · Decision Letter 0]

7 Jul 2023

PONE-D-23-05455

Epigenetic, psychological, and EEG changes after a 1-week retreat based on mindfulness and compassion for stress reduction: Study protocol of a cross-over randomized controlled trial

PLOS ONE

Dear Dr. Vazquez,

Thank you for submitting your manuscript to PLOS ONE. After careful consideration, we feel that it has merit but does not fully meet PLOS ONE’s publication criteria as it currently stands. Therefore, we invite you to submit a revised version of the manuscript that addresses the points raised during the review process.

We look forward to receiving your revised manuscript.

Kind regards,

Jan Christopher Cwik, Ph.D.

Academic Editor

PLOS ONE

Journal Requirements:

"This research has been partially funded by a grant from Ministry of Science (PID2019-108711GB-I00) to Carmelo Vazquez, and a Spanish Ministry of Science FPI predoctoral fellowship (PRE2020-092011) to Rosaria M. Zangri.The funders did not and will not have a role in study design, data collection and analysis, decision to publish, or preparation of the manuscript."

"CV: Ministry of Science PID2019-108711GB-I00

RMZ:Spanish Ministry of Science FPI predoctoral fellowship (PRE2020-092011)

GD: Mindfulness and Cognitive Science Chair of Complutense University of Madrid.

The funders did not and will not have a role in study design, data collection and analysis, decision to publish, or preparation of the manuscript."

"The authors have declared that no competing interests exist"

4. We note that you have referenced (Jinpa GT. Compassion Cultivation Training (CCT): Instructor’s manual. Stanford, CA: Unpublished manuscript; 2010) which has currently not yet been accepted for publication. Please remove this from your References and amend this to state in the body of your manuscript: (ie “Bewick et al. [Unpublished]”) as detailed online in our guide for authors

5. We note that you have stated that you will provide repository information for your data at acceptance. Should your manuscript be accepted for publication, we will hold it until you provide the relevant accession numbers or DOIs necessary to access your data. If you wish to make changes to your Data Availability statement, please describe these changes in your cover letter and we will update your Data Availability statement to reflect the information you provide

Additional Editor Comments 

I thank all reviewers for their time and effort in reading this manuscript. As you can see from the review below, both reviewers pointed out several mainly methodological aspects that must be considered in revising the manuscript.

Reviewers' comments:

Reviewer's Responses to Questions

**Comments to the Author**

1. Does the manuscript provide a valid rationale for the proposed study, with clearly identified and justified research questions?

Reviewer #1: Partly

Reviewer #2: Yes

2. Is the protocol technically sound and planned in a manner that will lead to a meaningful outcome and allow testing the stated hypotheses?

Reviewer #1: Partly

Reviewer #2: Partly

3. Is the methodology feasible and described in sufficient detail to allow the work to be replicable?

Reviewer #1: No

Reviewer #2: Yes

4. Have the authors described where all data underlying the findings will be made available when the study is complete?

Reviewer #1: Yes

Reviewer #2: Yes

5. Is the manuscript presented in an intelligible fashion and written in standard English?

Reviewer #1: Yes

Reviewer #2: Yes

6. Review Comments to the Author

You may also provide optional suggestions and comments to authors that they might find helpful in planning their study.

Reviewer #1: The manuscript entitled ‘Epigenetic, psychological, and EEG changes after a 1-week retreat based on mindfulness and compassion for stress reduction: Study protocol of a cross-over randomized controlled trial’

The manuscript could be improved.

Comments

Page 6, information on how the questionnaires will be administered here, although it was mentioned on Page 11 via Quatrics. Information on allocation concealment is to be mentioned and state blinding is impossible except for blinding for outcome measurement.

Page 7, the information on how the participants are led to the Nirakara-Lab and university’s official website offering the MBS and CCT programs (www.nirakara.org) is to be provided.

Page 8 Line 11, information on no ‘wash-out period’ is required before the subjects switch over to another intervention is to be mentioned.

Page 9 Line 6-8, more information on sample size calculation is to be provided e.g. outcome variable, 1 or 2-tailed test, type of subjects/group involved. The power is to be written as 95%. Why power 95% was chosen? The word ‘alpha standard error probability of’ requires revision.

Page 10 Line 16, information to be provided and if need be, followed by the statement ‘further details can be obtained from Table 2’. Avoid using referring to’ and leave the section empty without describing anything.

Page 11 -12, the proposed questionnaires used in the study are too many and may be taxing to the participant(s). How confident are the authors that the participants will eventually complete all the questionnaires and invite good-quality response or feedback?

Page 15 Line 5, the equivalent statistical test(s) is to be provided if missing data and attrition rates are a concern, as well as when the data is not missing at random (MAR). The effect size and 95% CI are to be indicated where applicable.

Page 15 Line 13, the version and publisher name of the statistical software is to be provided.

Page 16 Line 12-14, the sentence requires revision.

Page 16, Line 16-17, it would be good to state ‘the pattern and percentage of missing data is be explored prior to using the statistical test’. Binary logistic regression assumes data that is MCAR and MAR.

Page 22 Ensure the measures used are similar to the ones mentioned in Page 11-12 and other sections and remove the overlapping measures that capture the same thing.

Ensure the information in SPIRIT checklist are properly addressed and mentioned in the manuscript.

Some references did not conform to the journal format.

Reviewer #2: Summary:

This manuscript describes a study protocol of cross-over randomized controlled trial to study epigenetic, psychological, and EEG changes after a 1-week retreat based on mindfulness and compassion for stress reduction. The protocol is well-written, and the findings will provide unique insights in neurobiological and psychological mechanisms related to meditation and compassion. However, the current protocol has several issues that should be addressed.

Major comments:

1. Timepoints for measurements:

a. Epigenetic changes – There will be very limited-no variations in DNA methylation and gene expression from day 1 to day 4 and day 7. These changes are very slow and thus measuring epigenetic changes within 7 days is not informative. The article cited in line 15 stating that gene expression was modified after 8-day retreat had their measurements taking at 4 time points- 5-8 week before retreat, day of retreat before meditation, after retreat and 3 months after retreat. Similarly for reference 32-36, the study population and time points are different than those in this study. The authors might want to reconsider measuring epigenetic changes at day 1, day 7 and then at 3-month follow-up time to observe meaningful changes, however in doing so, the use of cross-over design may not be appropriate.

b. My thoughts on other outcomes are also similar that there might be very less variation in the measures from day 1 to day 4. The authors should reconsider the timepoints or provide justification using previous literature for the same.

c. If all the outcome measures are to have limited variation, then the cross-over study design may not be important.

d. The authors can consider other outcomes such as blood pressure and heart rate.

2. Sample size:

a. The sample size calculation is done using mean effect size of 0.37 which in the paper cited is for well-being outcomes. The authors need to recalculate the power and sample size using effect size of 0.78 which is for mindfulness outcomes.

b. As the authors are studying multiple outcomes, the sample size requirements will be different for each outcome. The effect size of a mindfulness outcome may not be comparable to other outcomes, and the authors need to consider this in sample size calculation.

3. The study will be conducted in a general population sample of healthy volunteer adults, but the authors do not provide any information about the demographic characteristics. Including such details would clarify the generalizability of the study findings. Also, the importance of conducting this retreat in general population should be justified.

Minor comments:

1. Please provide some information about the validity and reliability of the measures in general population.

2. Please provide a rationale for using a 3-month follow-up period, and why only self-report measures will be used.

3. The authors need to provide a brief summary of expected findings and implications to better understand the potential scientific contributions of this study.

7. PLOS authors have the option to publish the peer review history of their article (what does this mean?). If published, this will include your full peer review and any attached files.

Reviewer #1: No

Reviewer #2: No

---

## [Author Response · Author response to Decision Letter 0]

21 Aug 2023

Responses to the reviewers

Reviewer #1: The manuscript entitled ‘Epigenetic, psychological, and EEG changes after a 1-week retreat based on mindfulness and compassion for stress reduction: Study protocol of a cross-over randomized controlled trial’

The manuscript could be improved.

Comments

Page 6, information on how the questionnaires will be administered here, although it was mentioned on Page 11 via Quatrics. Information on allocation concealment is to be mentioned and state blinding is impossible except for blinding for outcome measurement.

RESPONSE. First of all, thank you for giving us the opportunity to improve the text of this design.

Although we indeed explained on p11 the administration procedure via Qualtrics, we explain now on p6-p7 that the administration of questionnaires will be done online. We now also provide information on allocation concealment.

Page 7, the information on how the participants are led to the Nirakara-Lab and university’s official website offering the MBS and CCT programs (www.nirakara.org) is to be provided.

RESPONSE. We now explain better the characteristics of this website (p. 7), which will be used as a main source of dissemination of the study. However, as we now also say, recruitment will be open to any interested individual who meets the conditions for participation in the study.

Page 8 Line 11, information on no ‘wash-out period’ is required before the subjects switch over to another intervention is to be mentioned.

RESPONSE. We now explicitly mention in the manuscript that there will be no wash-out period in the intervention (p. 8). Although this strategy is common in pharmacological treatments, such a wash-out period is not common in psychological interventions with similar designs (e.g., Geschwind et al., 2019). On the other hand, due to budgetary constraints, we cannot ask participants to return to the retreat site, days, weeks, or months after the first intervention. In addition, this wash-out could also introduce confounders that are difficult to control because they would occur outside the retreat (e.g., life stressors, variations in diet, differences in daily routines, etc.). For all of these reasons, we believe that a wash-out period is not an adequate strategy given the conditions of our study. 

Page 9 Line 6-8, more information on sample size calculation is to be provided e.g. outcome variable, 1 or 2-tailed test, type of subjects/group involved. The power is to be written as 95%. Why power 95% was chosen? The word ‘alpha standard error probability of’ requires revision.

RESPONSE. We thank the reviewer for this observation. The use of 95% power is common in studies of interventions in the field of psychology, although the minimum recommended in this field is 80% (https://doi.org/10.1525/collabra.28250). Our goal is to reduce the chances of Type II errors and we believe that this is an appropriate choice. We have now improved the description on sample size calculation in the manuscript (p. 9). 

Page 10 Line 16, information to be provided and if need be, followed by the statement ‘further details can be obtained from Table 2’. Avoid using referring to’ and leave the section empty without describing anything.

RESPONSE. This line has been removed from the Section and we include a footnote directing the reader to Table 2 for more details of the measurements and the timing of assessment.

Page 11 -12, the proposed questionnaires used in the study are too many and may be taxing to the participant(s). How confident are the authors that the participants will eventually complete all the questionnaires and invite good-quality response or feedback?

RESPONSE. Very similar protocols have been used previously in MBI research (e.g., Roca et al., 2019, 2021) and therefore we have full confidence that participants will complete the information. On the other hand, the total duration of the questionnaires does not exceed 35-40 minutes in total and, in the retreat conditions, completing psychological questionnaires at various times is perfectly adequate and feasible (e.g., Montero et al., 2016, https://doi.org/10.3389/fpsyg.2016.01935).

Page 15 Line 5, the equivalent statistical test(s) is to be provided if missing data and attrition rates are a concern, as well as when the data is not missing at random (MAR). The effect size and 95% CI are to be indicated where applicable.

RESPONSE. We have added more information on data analysis in the case of data not being missing at random (MNAR). In addition, we also explain that, in the case of questionnaires, the administration via Qualtrics, in which the participant is prevented from continuing if he/she has not answered all the items of a questionnaire, minimizes the possibility of having missing data in those instruments.

Page 15 Line 13, the version and publisher name of the statistical software is to be provided.

RESPONSE. We now provide this complete information

Page 16 Line 12-14, the sentence requires revision.

RESPONSE. Thank you. We have now revised this sentence to clarify the use of two-tailed tests.

Page 16, Line 16-17, it would be good to state ‘the pattern and percentage of missing data is be explored prior to using the statistical test’. Binary logistic regression assumes data that is MCAR and MAR.

RESPONSE. We now add this sentence in the revised text. 

Page 22 Ensure the measures used are similar to the ones mentioned in Page 11-12 and other sections and remove the overlapping measures that capture the same thing.

RESPONSE. We have double-checked that there is no overlap between the measurements and all of them are necessary for the study.

Ensure the information in SPIRIT checklist are properly addressed and mentioned in the manuscript.

RESPONSE. We have carefully reviewed the information that we already submitted with the original article, and which we do not know if the reviewer had access to it, so that everything is correctly reflected in the SPIRIT checklist and mentioned in the manuscript.

Some references did not conform to the journal format.

RESPONSE. Thanks for the observation, we have double-checked all the references and hope that everything is correct. In any case, in the original version of the manuscript and in this corrected version we have used the Mendeley plug-in for PlosONE.

Reviewer #2: Summary:

This manuscript describes a study protocol of cross-over randomized controlled trial to study epigenetic, psychological, and EEG changes after a 1-week retreat based on mindfulness and compassion for stress reduction. The protocol is well-written, and the findings will provide unique insights in neurobiological and psychological mechanisms related to meditation and compassion. However, the current protocol has several issues that should be addressed.

Major comments:

1. Timepoints for measurements:

a. Epigenetic changes – There will be very limited-no variations in DNA methylation and gene expression from day 1 to day 4 and day 7. These changes are very slow and thus measuring epigenetic changes within 7 days is not informative. The article cited in line 15 stating that gene expression was modified after 8-day retreat had their measurements taking at 4 time points- 5-8 week before retreat, day of retreat before meditation, after retreat and 3 months after retreat. Similarly for reference 32-36, the study population and time points are different than those in this study. The authors might want to reconsider measuring epigenetic changes at day 1, day 7 and then at 3-month follow-up time to observe meaningful changes, however in doing so, the use of cross-over design may not be appropriate.

RESPONSE. Thank you for the opportunity to review the article and recognize the potential advances that our study may bring to the field of compassion and meditation.

RESPONSE. As we indicated to the first reviewer, this is a study with a very limited budget and while we understand that it would be ideal to include measurements of epigenetic changes with a longer time window, we can only do it during the 7-day retreat. In addition, relevant long-term follow-up measurements of epigenetic and transcriptomic changes might be very challenging, as they may be biased by many external factors difficult to control after the retreat. For instance, high variability in participant lifestyle, including diet, meditation practice routine, or even health problems and infections, may influence epigenetic results, not to mention the difficulties in logistics, as our participants may come from distal places of residence and will be in most cases unable to return to the place of the retreat 3 or 12 months after. However, previous evidence suggests that it is possible to find changes associated with intensive training in a limited period of time (i.e., 3-7 days). In fact, although in the study by Chandran et al. that the reviewer was mentioning (ref 31 of the submitted version) transcriptional changes were investigated in 4-time points, most of the differentially expressed genes were detected immediately after 8-days retreat (44% of differentially expressed genes were exclusive to T3, after retreat). These changes were related mostly to the immune response, immune system, and signaling pathways, which is one of the most consistent changes associated with mind-body interventions observed across different studies (independently of the duration of the intervention). In this context, it should be highlighted that Chandran et al. observed a reversion of most of the genes related to immune-response related in T4, during their 3-month follow-up, which may be due to external factors affecting immediate retreat-related transcriptional effect.

Furthermore, although we agree with the reviewer that the epigenetic changes might accumulate throughout extended periods of time, it is also known that epigenetic alterations might actually be very dynamic upon the activity of different stimuli. Even in an 8-hour intensive mindfulness-based retreat, significant transcriptional (Kaliman et al, ref 36 of the submitted version) as well as DNA methylation changes were observed in blood samples, and affected in genes related to important cellular pathways (such as fatty acid metabolism, DNA repair, and chromatin remodeling, among others)(Chaix et al., ref 35 of the submitted version). Furthermore, differentially methylated CpGs were enriched in binding sites of particular transcription factors, involved in immune response and inflammation (ref 35). These findings suggest that the changes observed are not random, and seem to be induced by the meditation practice, even after only one day of intervention.

Although we do not know a priori whether we will be able to detect highly relevant epigenetic changes in the blood of the participants of our retreat, there is sufficient published evidence to expect some degree of epigenetic modulation after short periods of practice, supporting thus our experimental design. 

In any case, we thank the reviewer for this note of caution and include, in the revised version of the manuscript, some additional information in support of the possibility of finding changes in a brief intervention as planned in the present design.

b. My thoughts on other outcomes are also similar that there might be very less variation in the measures from day 1 to day 4. The authors should reconsider the timepoints or provide justification using previous literature for the same.

RESPONSE. Since the retreat is to last one week, which is the most common duration for meditation retreats (Khoury et al., 2017), we must include the measures in that short period of time. However, as the contents of the meditation program are intense (the standard contents of the MBSR and CCT programs are offered in an integral way), we believe that it is possible to find significant effects associated with these intensive interventions. We understand that it is a risk not to find differences in such a short period of time, but it seems to us that it is plausible to find significant differences in the direction indicated in the hypotheses. We now add this clarification in discussing the limitations of the study.

c. If all the outcome measures are to have limited variation, then the cross-over study design may not be important.

RESPONSE. We are not sure that the outcome measures necessarily have limited variation. Perhaps the greatest limitation in variability comes from the fact that participants are drawn from the general population but as explained in the protocol we will exclude candidates with significant medical and psychological conditions, so the sample will be of mostly healthy individuals. In any case, we expect that the cross-over design will allow us, even with all the limitations that the reviewer indicates and that we also indicate in the text, to find differences in the order of application of the interventions. The interest of this type of design is precisely to analyze the impact of the sequence of interventions, apart from their additive effects. In this sense, we believe that the cross-over design continues to be of interest in our study. The question of whether it is more beneficial to begin with mindfulness practice or compassion practice is a pivotal inquiry in the field of meditation-based interventions. Given that our study seeks to determine if there is a benefit in starting with one practice over the other, a cross-over design allows for a direct evaluation of the introduction sequence of the practices and their ensuing effects.

d. The authors can consider other outcomes such as blood pressure and heart rate.

RESPONSE. Thank you very much for this observation. After reading your observation, we have decided to include measures of heart rate and heart rate variability (HRV), and respiratory rate for which our team has accredited experience.

2. Sample size:

a. The sample size calculation is done using mean effect size of 0.37 which in the paper cited is for well-being outcomes. The authors need to recalculate the power and sample size using effect size of 0.78 which is for mindfulness outcomes.

RESPONSE. Thank you for this interesting observation. We explicitly decided to use the effect size of well-being measures for two reasons. First, we are comparing mindfulness training (MBSR) and compassion training (CCT); we believed that using the mindfulness effect size, in this comparison, would be a biased choice toward one of the interventions (i.e., MBSR). On the other hand, we have healthy participants for our study, and it seems more appropriate to use measures that capitalize on aspects of psychological well-being rather than on clinical measures. Of note, the choice of this effect size implies having a larger sample of participants, all other conditions being equal.

b. As the authors are studying multiple outcomes, the sample size requirements will be different for each outcome. The effect size of a mindfulness outcome may not be comparable to other outcomes, and the authors need to consider this in sample size calculation.

RESPONSE. We agree with this comment. It is true that in a complex design with multiple measures, there is no single indication of the effect size required for each measure. However, we have used a very conservative strategy since our sample size calculations are based on studies with psychological measures. These studies have much larger sample requirements than studies in neuroscience (Marek et al. (2002). Nature, https://doi.org/10.1038/s41586-022-04492-9; Szucs & Ioannidis (2020), Neuroimage, https://doi.org/10.1016/j.neuroimage.2020.117164). This decision may imply that biological measures may be overpowered in terms of sample size, but it is also possible that in these measures there may be more missing data due to difficulties in recording. In any case, we will take this important observation into account in the conclusions we draw from the data in all future analyses.

3. The study will be conducted in a general population sample of healthy volunteer adults, but the authors do not provide any information about the demographic characteristics. Including such details would clarify the generalizability of the study findings. Also, the importance of conducting this retreat in the general population should be justified.

RESPONSE. Demographic data will be provided at the end of the study. Our plan is, as we now make it clear in the manuscript, that the composition of men and women in each of the two groups in the design will be equal, so randomization will take this matching factor into account. We now also add some additional information on the demographic variables that we will collect in the study. 

Given the increasing popularity of mindfulness and compassion practices and research in the general population, it is crucial to understand their effects on individuals who do not necessarily have clinical problems but seek personal growth, stress reduction, or well-being enhancement. We have made significant efforts in reinforcing our exclusion criteria to ensure we select only a healthy population. This rigorous approach ensures the integrity of our findings, free from potential confounders related to health conditions. Our intention is to assess the impact of these retreats on a healthy population, aiming to provide tools that enhance well-being and contribute to the prevention of mental health issues. By understanding the effects on a healthy cohort, we can offer evidence-based guidance on how these practices can serve as preventative measures for the broader community, potentially reducing the onset of mental health conditions. We now include these reflections in the Discussion section.

Minor comments:

1. Please provide some information about the validity and reliability of the measures in general population.

RESPONSE. As we mention now in the manuscript, the criterion for the selection of measures for the design of this study was precisely to use measures that have been commonly used in the general population. They are not clinical measures and have had a very common use in meditation studies. In fact, we indicate a general reference to a study (Roca et al., 2022) that has used a very similar set of measures. The measures are well-known in the psychological literature and are perhaps the most standard self-report tools for assessing the psychological constructs we wish to measure (positive and negative emotions, life satisfaction, mindfulness, compassion, etc.). Including data on each and every measure in terms of reliability and validity, which are common measures in psychological research, maybe too much information for a schematic paper focused on design. However, if the reviewer deems it necessary to include this detailed information, we can include reliability and validity indices for each measure.

2. Please provide a rationale for using a 3-month follow-up period, and why only self-report measures will be used.

RESPONSE. Thank you for this comment. After having read it, we believe it is more appropriate to extend the follow-up and also to make an evaluation 1 year after the intervention, which we now include in the protocol. This follow-up, at 3 months and 12 months, will be done only with self-report measures due to logistical and budget constraints. Biological testing requires setting up laboratories for blood analysis, psychophysiological measures, etc. All of this would require participants to return again to the laboratories from their places of residence. While that would be ideal, it is beyond the limited resources we have for this research.

3. The authors need to provide a brief summary of expected findings and implications to better understand the potential scientific contributions of this study.

RESPONSE. We now include that brief summary of expected findings and implications.

---

## [Decision Letter · Decision Letter 1]

25 Oct 2023

Epigenetic, psychological, and EEG changes after a 1-week retreat based on mindfulness and compassion for stress reduction in healthy adults: Study protocol of a cross-over randomized controlled trial

PONE-D-23-05455R1

Dear Dr. Vazquez,

We’re pleased to inform you that your manuscript has been judged scientifically suitable for publication and will be formally accepted for publication once it meets all outstanding technical requirements.

Kind regards,

Jan Christopher Cwik, Ph.D.

Academic Editor

PLOS ONE

Additional Editor Comments (optional):

Reviewers' comments:

Reviewer's Responses to Questions

**Comments to the Author**

1. Does the manuscript provide a valid rationale for the proposed study, with clearly identified and justified research questions?

Reviewer #1: Yes

Reviewer #2: Yes

2. Is the protocol technically sound and planned in a manner that will lead to a meaningful outcome and allow testing the stated hypotheses?

Reviewer #1: Partly

Reviewer #2: Yes

3. Is the methodology feasible and described in sufficient detail to allow the work to be replicable?

Reviewer #1: Yes

Reviewer #2: Yes

4. Have the authors described where all data underlying the findings will be made available when the study is complete?

Reviewer #1: Yes

Reviewer #2: Yes

5. Is the manuscript presented in an intelligible fashion and written in standard English?

Reviewer #1: Yes

Reviewer #2: Yes

6. Review Comments to the Author

You may also provide optional suggestions and comments to authors that they might find helpful in planning their study.

Reviewer #1: The authors have put in great effort to address the comments and revised the manuscript.

I have no further comments.

Reviewer #2: The authors have addressed the concerns in their responses and made necessary changes in the revised manuscript to improve the content.

7. PLOS authors have the option to publish the peer review history of their article (what does this mean?). If published, this will include your full peer review and any attached files.

Reviewer #1: No

Reviewer #2: No

---

## [Editor Report · Acceptance letter]

7 Nov 2023

PONE-D-23-05455R1 

Epigenetic, psychological, and EEG changes after a 1-week retreat based on mindfulness and compassion for stress reduction in healthy adults: Study protocol of a cross-over randomized controlled trial 

Dear Dr. Vázquez:

I'm pleased to inform you that your manuscript has been deemed suitable for publication in PLOS ONE. Congratulations! Your manuscript is now with our production department. 

Kind regards, 

on behalf of

Dr. Jan Christopher Cwik 

Academic Editor

PLOS ONE